# Concept-Based Steering of LLMs for Conditional Molecular Generation

## Abstract

Generating valid, unique, and high-fidelity molecules while precisely control-
ling for multiple properties simultaneously remains challenging. While prior
works with LLMs have achieved success by fine-tuning language models on novel
molecular corpora, they remain limited in scope. Real-world applications require
generating molecules from unseen property distributions, a task that remains chal-
lenging for fine-tuned models. To this end, we present Concept-based Activation
STeering (CAST), the first approach to apply activation steering to directly edit
a model's internal representation for conditional molecular generation. CAST
offers a lightweight, flexible alternative to fine-tuning by computing property-
conditioned steering vectors via a concept network that does not require retrain-
ing the LLM. Through extensive experiments on datasets such as Therapeutics
Data Commons, we show that CAST consistently outperforms existing methods
on both in-distribution and out-of-distribution conditional generation tasks. We
also conduct comprehensive ablation studies to highlight the extent of control our
concept-guided steering provides on the molecules generated by the LLM.

## 1 Introduction

The ability to efficiently generate valid, unique, and novel molecules with targeted properties has
become crucial for accelerating drug discovery and the development of advanced materials (Mer-
chant et al., 2023; M. Bran et al., 2024; Ma et al., 2024; Wang et al., 2025). This challenge has
spurred increasing interest in the task of conditional molecular generation, which enables the de-
sign of molecules optimized for specific characteristics, such as enhanced drug-likeness (QED)
and favorable solubility (LogP), both in the setting of single and more challenging multi-property
generation. Furthermore, this task entails complications in generating diverse molecular structures
that satisfy multiple property constraints simultaneously (Jang et al., 2025), especially when target
properties fall outside the distribution of the training data.

Recently, large language models (LLMs) trained on text-based molecular representations, such as
SMILES strings, have shown promising capabilities in generating valid and high-fidelity molecules.
In particular, tags representing molecular properties have been incorporated into some LLMs to
enable conditional generation (Guevorguian et al., 2024). For better performance, traditionally,
two main paradigms have evolved: supervised fine-tuning (SFT) and reinforcement learning (RL).
Within the SFT category, Fan et al. (2025) combine learnable numerical and text embeddings during
fine-tuning, improving the model's fidelity. Other research, such as Li et al. (2024); Lin et al. (2025);
Li et al. (2025), proposes methods that include dynamic context integration and multi-step instruc-
tion tuning. However, these SFT approaches are resource-intensive and lack flexibility, as each new
property distribution or objective demands costly retraining. RL-based methods are more adaptable
since they optimize behavior directly from feedback, but they suffer from the inherent challenge of
designing suitable reward functions and accessing accurate oracles for learning. Some works also
combine the two approaches for better results. For example, Cavanagh et al. (2025) leverages DPO
after SFT to improve alignment to specified property values. Similarly, Jang et al. (2025) optimizes
the structural diversity of the generated molecules with a tailored reward function. However, these
methods are typically computationally expensive, limiting their flexibility and practical applicability.

Alternatively, *activation steering* has emerged as a promising paradigm for model alignment (Turner
et al., 2024; Zou et al., 2025). By directly editing hidden-layer activations in pretrained models, ac-

tivation steering can guide model outputs toward desired behaviors without altering model weights. Besides being more efficient than fine-tuning, this method has already demonstrated improved interpretability, flexibility, and precision in various NLP tasks such as enhancing trustworthiness and mitigating biases (Zou et al., 2025; Bayat et al., 2025). In conditional molecular generation, given the dominance of string-based molecular representations such as SMILES and the remarkable capabilities of pretrained LLMs, activation steering presents a compelling new direction. While traditional activation steering, common in NLP applications, shows promise, it does present some limitations, such as being largely dependent on manually crafted contrastive prompts and the need to tune the steering strength, which can limit its effectiveness across diverse samples.

In this paper, we first explore the feasibility of activation steering to conditional molecular generation using LLMs. Building on our observations, we further propose a novel steering approach, Concept-based Activation STeering (CAST) that leverages a Concept Bottleneck Model (CBM) (Koh et al., 2020) to automatically compute property-conditioned steering vectors with appropriate steering strength calculated dynamically during inference. This approach preserves the interpretability advantages of activation steering while addressing its inherent shortcomings.

The main contributions of our paper are as follow:

- We evaluate activation steering as a promising alternate method for conditional molecular generation with LLMs. We further identify and address its key limitations, including reliance on manually crafted contrastive prompts, fixed steering magnitudes, and manual construction of steering vectors.
- We propose Concept-Based Activation Steering (CAST), which incorporates a concept bottleneck model to compute property-conditioned steering vectors.
- We demonstrate CAST's strong and robust performance through comprehensive multi-property conditional generation experiments on both in-distribution and out-of-distribution settings. Finally, we present extensive qualitative and quantitative studies to establish the high quality and extent of control over properties of molecules generated by CAST.

## 2 RELATED WORK

### 2.1 LLMs FOR MOLECULAR GENERATION

Advances in LLMs have opened new directions for leveraging diverse text-based representations such as SMILES and SELFIES in scientific applications, particularly molecular generation. While early works (Bagal et al., 2022; Edwards et al., 2022) demonstrated success in training smaller Transformer-based models; the field has increasingly shifted toward more capable LLMs. Recent studies have shown that incorporating domain-specific chemical knowledge through instruction-tuning greatly enhances molecular generation performance (Yu et al., 2024; Zhang et al., 2024; Fang et al., 2024). However, supervised fine-tuning methods face inherent limitations in generating diverse molecular structures with desired properties (Jang et al., 2025). Other works involving LLMs have also explored preference tuning techniques, including reinforcement learning and offline methods, to incorporate feedback in their generation. Notable examples include Cavanagh et al. (2025) and Jang et al. (2025), who employ RL-based approaches to improve alignment with specific property constraints and structural diversity. Despite these advances, achieving precise conditional control over multiple molecular characteristics simultaneously remains a significant challenge.

### 2.2 ACTIVATION STEERING

Activation steering, first introduced in the ActAdd paper (Turner et al., 2024), has gained attention as an alternative to prompt engineering and fine-tuning for aligning LLMs' behaviors. Unlike model editing techniques that directly modify model weights, activation steering constructs steering vectors that are added to the residual stream of a frozen LLM, influencing the output generation process. The most common approach for computing steering vectors involves using contrastive prompts $(p_+, p_-)$ that elicit desired and undesired behaviors, respectively. The steering vector is then derived by taking the difference between their corresponding activation vectors $(h_+^l, h_-^l)$ at intermediate layer $l$ (Refer to Fig. 1 in Turner et al. (2024)). Alternative methods, such as the approach proposed by Zou et al. (2025), compute steering vectors by training a linear classifier on the activation pairs $(h_+^l, h_-^l)$ and

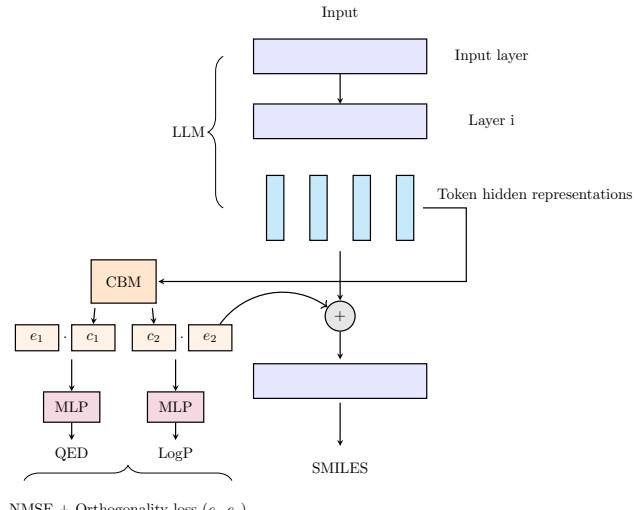

Figure 1: Architecture of the Concept-based Activation STeering (CAST) framework.

extracting the normal vector to the decision boundary as the steering direction. During inference, these steering vectors are applied by adding them to the model's hidden activations in the forward pass (Turner et al., 2024), effectively manipulating the model's internal representations toward the desired behavioral patterns without requiring parameter updates. Moreover, most works such as Zou et al. (2025); Bayat et al. (2025); Panickssery et al. (2024); Lee et al. (2025) have primarily focused on behavioral alignment tasks, such as enhancing refusal, truthfulness, honesty, or mitigating various forms of bias. To our knowledge, this work represents the first attempt to apply activation steering techniques to molecular generation.

## 3 CAST: CONCEPT-BASED ACTIVATION STEERING

We propose the Concept-based Activation STeering (CAST) framework for aligning molecular generation language models to any arbitrary target property distribution, without requiring computationally intensive fine-tuning or post-training methods. CAST is a novel approach for generating steering vectors explicitly conditioned on interpretable input properties. Prior steering methods often rely heavily on the quality of hand-crafted contrastive sets of prompts and are restricted to manually extracting a single steering vector per property of interest. CAST, however, overcomes these limitations by training a concept bottleneck model to automatically and adaptively compute steering vectors grounded in property values. In our work, we view the different properties of interest as *concepts* and will use the terms interchangeably. This section discusses the input setup, the proposed CBM architecture, training, and inference procedures.

**Input Setup** We adopt the structured input prompt from Guevorguian et al. (2024) for our molecular generation task. Each molecular input sequence is composed of special tokens denoting molecular properties, followed by the SMILES string. Specifically, each input sequence begins with special property tags indicating desired target properties, i.e., `[QED]0.8[/QED][START_SMILES]`. The model generation task involves completing this input sequence by generating the corresponding SMILES representation, ending with an `[END_SMILES]` token. This format ensures that the model is explicitly conditioned on the desired molecular property values during generation.

### 3.1 ARCHITECTURE

The proposed architecture of our Concept-based Activation STeering framework is shown in Figure 1. The framework operates on top of a pre-trained decoder-only large language model that we aim to steer. Given an input sequence $x = \{x_1, x_2, ..., x_n\}$, the LLM processes the $n$ tokens through $L$ transformer layers, producing hidden states $\mathbf{H}^{(l)} \in \mathbb{R}^{n \times d}$ at each layer, where $d$ is the hidden

size. We select a specific intermediate layer $l$ and extract the hidden representations $\mathbf{h} \in \mathbb{R}^{n \times d}$. Intermediate layers are generally the most suitable for steering, as they retain rich semantic information Lucchetti & Guha (2024), and we use the midpoint layer as a practical choice. Considering the decoder-only architecture of the LLMs, we only use the hidden representation corresponding to the last token, denoted as $h_n$, as input to the CBM.

**Concept Bottleneck Module** In order to derive steering vectors from interpretable molecular properties, we follow a concept bottleneck design. Similar to Ismail et al. (2024), we design the CBM to receive the extracted hidden representations $\mathbf{h}_n \in \mathbb{R}^d$ from the LLM as input and learns a mapping function $f : \mathbb{R}^d \to \mathbb{R}^k$ that transforms the internal representations to scalar concept values $c_i$ for each target molecular property $\mathbf{i} \in \{1, 2, ..., k\}$, where $k$ is the number of target properties. To ensure concept-specific directions within the latent activation space, each concept value $c_i$ is associated with a learnable embedding vector $\mathbf{e}_i \in \mathbb{R}^d$. These embeddings encode fixed directional bases for each property, learned during training. Dynamic steering is then achieved through the concept values $c_i$ that modulate the magnitude of each concept-specific direction. The final representation $\mathbf{z}_i$, conditioned on target molecular properties, is computed through the product:

$$\mathbf{z}_i = c_i \mathbf{e}_i \tag{1}$$

**Property Predictor Module** To train the embeddings that define our steering vector, we introduce a feedback loop that allows backpropagation of a loss signal through the CBM. Each resulting representation $\mathbf{z}_i$ is further processed by a dedicated multi-layer perceptron (MLP) to reconstruct the ground-truth molecular property values provided in the prompt. The training objective minimizes the Normalized Mean Squared Error (NMSE) loss between the predicted values ($\hat{y}_i$) and their actual input values ($y_i$):

$$\mathcal{L}_{\text{NMSE}} = \frac{1}{k} \sum_{i=1}^{k} \frac{(y_i - \hat{y}_i)^2}{\text{Var}(\{y_j\}_{j=1}^{k})}. \tag{2}$$

Additionally, an orthogonality loss is imposed between embedding vectors $\mathbf{e}_i$, promoting disentangled and interpretable property representations:

$$\mathcal{L}_{\text{total}} = \mathcal{L}_{\text{NMSE}} + \lambda \mathcal{L}_{\text{orth}}. \tag{3}$$

Importantly, backpropagation is performed only through the CBM and property MLPs while keeping the underlying LLM parameters frozen. We use $\lambda = 1$ in our experiments.

**Activation Steering** At inference, we can compute the steering vector $\mathbf{z}$ by summing all property-specific vectors $\mathbf{z} = \sum_i \mathbf{z}_i$. This aggregated steering vector can then be simply added to the LLM's hidden activations at the chosen intermediate layer as follows:

$$\tilde{\mathbf{h}} = \mathbf{h} + \mathbf{z}. \tag{4}$$

The modified hidden activations $\tilde{\mathbf{h}}$ then propagate through the remaining transformer layers, steering the generation to produce a SMILES string that exhibits the desired molecular properties. While the model autoregressively generates the output sequence token by token, we apply this activation steering at each generation step by adding $\mathbf{z}$ to the hidden representation at the selected layer. This ensures that the property-driven guidance is maintained throughout the entire decoding process, dynamically aligning the model's outputs with the input values during the full sequence generation. Note that during the generation process, $\mathbf{z}$ remains constant, ensuring that generated tokens do not influence the steering vector.

## 4 EXPERIMENTS

To extensively assess our proposed method, we perform multi-property conditional generation experiments across both in-distribution settings, where target property values correspond to common property targets seen in the training data, and out-of-distribution settings, where targets fall outside these typical regions. We also conduct qualitative studies to further understand the nuances of the CAST framework.

**Datasets** For in-distribution evaluation, we use three datasets from the Therapeutics Data Commons (TDC) (Huang et al., 2021) molecular generation task, comprising the ZINC (Sterling & Irwin,

Table 1: In-Distribution results on the TDC Datasets. Performance is reported for QED MAE, LogP MAE, and Validity (%). Lower MAE and higher validity indicate better performance. We also include # Trainable parameters for each method. **Bolded** results indicate the best results without including fine-tuning methods, as we use them for reference.

| Base LLM | Method | #Train. params. | Zinc | | | Moses | | | ChemBL | | |
|---|---|---|---|---|---|---|---|---|---|---|---|
| | | | QED MAE ($\downarrow$) | LogP MAE ($\downarrow$) | Validity ($\uparrow$) | QED MAE ($\downarrow$) | LogP MAE ($\downarrow$) | Validity ($\uparrow$) | QED MAE ($\downarrow$) | LogP MAE ($\downarrow$) | Validity ($\uparrow$) |
| **Chemlactica-125M** | Baseline | 0 | 0.086 | **0.472** | 99.7% | 0.126 | 1.342 | 99.5% | **0.185** | 2.015 | 99.3% |
| | +ActAdd | 0 | 0.141 | 2.955 | 93.3% | 0.175 | 2.097 | 93.3% | 0.348 | 9.079 | 86.8% |
| | +CAST | 7M | **0.075** | 0.611 | **100%** | **0.078** | **1.232** | **100%** | 0.226 | **1.738** | **99.5%** |
| | Full SFT | 125M | 0.025 | 0.127 | 100% | 0.097 | 1.446 | 100% | 0.202 | 1.747 | 99.9% |
| | LoRA SFT | 78M | 0.147 | 1.571 | 99.7% | 0.104 | 1.038 | 100% | 0.249 | 2.221 | 97.7% |
| **Chemlactica-1.3B** | Baseline | 0 | 0.063 | 0.391 | 97.3% | 0.085 | 1.461 | 99.9% | **0.182** | 1.793 | 92.0% |
| | +ActAdd | 0 | 0.126 | 2.119 | 65.8% | 0.115 | 1.483 | 77.4% | 0.394 | 2.229 | 57.8% |
| | +CAST | 50M | **0.054** | **0.386** | **100%** | **0.079** | **1.420** | **100%** | 0.221 | **1.571** | **99.2%** |
| | Full SFT | 1.3B | 0.030 | 0.171 | 100% | 0.107 | 1.471 | 100% | 0.176 | 1.604 | 99.8% |
| | LoRA SFT | 210M | 0.146 | 1.578 | 100% | 0.107 | 1.038 | 100% | 0.251 | 2.314 | 99.3% |
| **Chemma-2B** | Baseline | 0 | 0.231 | 8.824 | **100%** | 0.221 | 1.456 | **100%** | 0.232 | 23.928 | **96.9%** |
| | +ActAdd | 0 | 0.393 | 12.446 | 62.7% | 0.452 | 14.868 | 58.0% | 0.299 | 15.890 | 75.4% |
| | +CAST | 50M | **0.084** | **2.780** | 98.7% | **0.073** | **1.372** | 99.9% | **0.230** | **1.720** | 92.1% |
| | Full SFT | 2B | 0.020 | 0.125 | 99.9% | 0.099 | 1.459 | 99.9% | 0.176 | 1.604 | 99.8% |
| | LoRA SFT | 1B | 0.233 | 2.170 | 99.2% | 0.227 | 1.820 | 99.6% | 0.281 | 4.090 | 98.1% |
| **Qwen3-4B** | Baseline | 0 | 0.396 | 2.407 | 55.5% | 0.204 | 2.152 | 63.5% | 0.466 | 3.628 | 42.5% |
| | +CAST | 85M | **0.370** | **2.275** | **59.6%** | **0.228** | **1.911** | **70.9%** | **0.418** | **2.936** | **91.0%** |

2015), MOSES (Polykovskiy et al., 2020), and ChemBL (Mendez et al., 2018). We specifically chose TDC because it contains well-characterized, widely studied molecules with property distributions that are representative of real-world small-molecule drug discovery, minimizing the risk of out-of-distribution effects. To adapt these datasets for the conditional generation task, we use *RDKit* (Landrum et al., 2025) to compute ground truth values of QED and LogP, two continuous real-valued numerical properties. These datasets serve as the primary training and evaluation data for CAST.

For out-of-distribution experiments, we follow the setup defined in Jolicoeur-Martineau et al. (2024) to condition on specific single known out-of-distribution molecular properties. We also extend the methodology to include the combinations of out-of-distribution properties. Finally, we also evaluate on Conjugated-xTB (Jolicoeur-Martineau et al., 2025), a molecular dataset composed of organic $\pi$-conjugated molecules with out-of-distribution property values. We chose this dataset because these molecules tend to have structures significantly different from the molecules present in popular datasets like TDC. This enables us to evaluate "true" out-of-distribution performance better, as the LLM is less likely to have seen such structures in pretraining.

**Models** For all our experiments, we use the tag-based LLMs, Chemlactica-125M, Chemlactica-1.3B, and Chemma-2B (Guevorguian et al., 2024) and steer at layers 6, 12 and 9 respectively (approximately the halfway point of each model). To also test the generalizability of our method, we include experiments with a general-purpose LLM, Qwen3-4B (Yang et al., 2025) steered at layer 18. In addition, both the concept bottleneck module and property predictor modules trained for each of these models were composed of two intermediate layers of dimensions $4d$ and $2d$, where $d$ is the dimension of the hidden representation. Both modules were trained using the AdamW optimizer on a single NVIDIA A100 40GB GPU, with early stopping employed based on validation loss to prevent overfitting. For a holistic evaluation, we also compare against full supervised fine-tuning (SFT) and low-rank (LoRA) (Hu et al., 2022) fine-tuning versions of the base models. Note that in all tables, **bolded** results indicate the best results without including fine-tuning methods, as we use them for reference.

**In-distribution Metrics** We evaluate the quality and controllability of generated molecules using a suite of quantitative metrics that collectively assess property alignment, chemical validity, and molecular novelty. To evaluate fidelity with respect to input target properties, we mainly use the **Mean Absolute Error** (MAE). For a set of $N$ generated molecules, where each molecule $i$ is conditioned on a target property value $y_i$ and the resulting molecule has property $\hat{y}_i$, the MAE is:

$$\text{MAE} = \frac{1}{N} \sum_{i=1}^{N} |\hat{y}_i - y_i| \tag{5}$$

This metric quantifies the average deviation between each generated molecule's predicted property and its ground truth target, capturing overall conditional accuracy across diverse target values.

Table 2: Single Property OOD setting - Best100MAE and GenEff with QED and LogP targets. Note that it is impossible to obtain a QED value $> 1$; therefore, at best, the MAE can be 0.2861.

| Base LLM | Method | Single Property - Best100MAE ($\downarrow$) | | | | Single Property - GenEff(%) ($\uparrow$) | | | |
| | | QED | | LogP | | QED | | LogP | |
| | | 0.1778 | 1.2861 | -3.2810 | 8.1940 | 0.1778 | 1.2861 | -3.2810 | 8.1940 |
|---|---|---|---|---|---|---|---|---|---|
| **Chemlactica-125M** | Baseline | 0.0443 | 0.780 | 0.147 | 0.203 | 80.5 | 84.6 | 98.1 | 81.2 |
| | +CAST | **0.0231** | **0.413** | **0.108** | **0.079** | **92.5** | **95.5** | **98.8** | **97.9** |
| | Full SFT | 0.0040 | 0.698 | 0.0474 | 4.102 | 96.3 | 94.0 | 74.6 | 99.0 |
| | LoRA SFT | 0.0062 | 0.581 | 0.418 | 2.766 | 88.2 | 90.3 | 72.6 | 77.1 |
| **Chemlactica-1.3B** | Baseline | 0.0109 | 0.484 | 0.152 | 0.129 | 87.8 | 96.6 | 97.4 | 96.5 |
| | +CAST | **0.0097** | **0.467** | **0.092** | **0.061** | **88.5** | **98.4** | **98.2** | **99.2** |
| | Full SFT | 0.0234 | 0.550 | 0.200 | 5.625 | 75.1 | 79.9 | 82.1 | 97.9 |
| | LoRA SFT | 0.0044 | 0.487 | 2.494 | 2.407 | 97.2 | 99.0 | 98.8 | 97.4 |
| **Chemma-2B** | Baseline | 0.0106 | **0.420** | **0.139** | 0.193 | 92.5 | 98.6 | 96.7 | 90.2 |
| | +CAST | **0.0080** | 0.425 | 0.182 | **0.122** | **93.6** | 98.6 | **98.3** | **95.2** |
| | Full SFT | 0.0051 | 0.613 | 0.160 | 5.206 | 94.5 | 96.7 | 88.9 | 99.1 |
| | LoRA SFT | 0.0051 | 0.464 | 2.372 | 2.028 | 90.7 | 96.4 | 98.4 | 98.7 |

In addition to these metrics, we also use **validity** to quantify the quality of the generated molecules. Validity is defined as the proportion of generated molecules that correspond to chemically valid structures according to RDKit:

$$\text{Validity} = \frac{1}{N} \sum_{i=1}^{N} \mathbb{I} \left[ \text{mol}_i \text{ is valid} \right] \tag{6}$$

where $\mathbb{I}[\cdot]$ is the indicator function, and $\text{mol}_i$ denotes the i-th generated molecule.

**Out-of-Distribution Metrics** We first evaluate fidelity using **Best100MAE** where we select the 100 molecules with the smallest absolute property errors from $N$ generated candidates and compute their average.

In addition to validity, We further assess the novelty of the generation process using **generative efficiency** (GenEff). Generative efficiency is then defined as the probability of satisfying validity, uniqueness, and novelty:

$$\text{GenEff} = \frac{1}{N} \sum_{i=1}^{N} \mathbb{I} \left[ \text{mol}_i \text{ is valid, unique and novel} \right] \tag{7}$$

## 4.1 HOW WELL DOES CAST PERFORM ON IN-DISTRIBUTION DATA?

We evaluate CAST on three base models across the ZINC, MOSES, and ChEMBL molecular datasets, using $N = 2000$ randomly sampled QED and LogP property combinations per dataset. In addition to our base LLMs' vanilla, full SFT, and LoRA SFT baselines, we include the ActAdd method (Turner et al., 2024), a widely used activation steering approach, to directly compare its effectiveness with our proposed method. *As shown in Table 1, CAST consistently achieves lower QED and LogP mean absolute errors (MAEs) compared to both base model and ActAdd, demonstrating more precise and reliable control over generated molecular properties.* For instance, on Chemma-2B with ZINC, CAST attains a QED MAE of 0.084 and a LogP MAE of 2.780, representing over a 50% relative improvement in both properties MAE compared to the base model (QED MAE of 0.231 and LogP MAE of 8.824), while maintaining a competitive validity of 98.7%. We observe a similar pattern of improvements across the other datasets and models. Particularly, the performance gains from CAST are more pronounced as model size increases, suggesting that larger models with richer internal representations enable more effective activation steering. Importantly, CAST maintains high validity across all datasets (all above 90%) and consistently outperforms ActAdd in this regard, indicating that CAST can steer adequately toward more chemically meaningful regions. We also observe that ActAdd does not improve conditional generation, as we see an increase in MAEs across all datasets and models. As shown in Table 1, CAST consistently improves performance on

Table 3: Multiple OOD property combination: QED Target 0.1778, LogP Target 8.1940.

| Base LLM | Method | Best100MAE (↓) QED | Best100MAE (↓) LogP | GenEff (%) (↑) |
|---|---|---|---|---|
| Chemlactica-125M | Baseline | 0.0379 | 2.452 | 71.7 |
| | +CAST | **0.0182** | **0.633** | **90.7** |
| | Full SFT | 0.0057 | 2.163 | 96.7 |
| | LoRA SFT | 0.0031 | 0.223 | 86.7 |
| Chemlactica-1.3B | Baseline | 0.0104 | 0.432 | 86.4 |
| | +CAST | **0.0097** | **0.353** | **90.4** |
| | Full SFT | 0.0274 | 4.373 | 93.7 |
| | LoRA SFT | 0.0025 | 0.062 | 92.6 |
| Chemma-2B | Baseline | 0.0137 | 0.992 | 55.3 |
| | +CAST | **0.0124** | **0.863** | **59.6** |
| | Full SFT | 0.0034 | 3.752 | 94.4 |
| | LoRA SFT | 0.0041 | 0.154 | 88.4 |

Table 4: Performance on OOD Conjugated-xTB dataset.

| Base LLM | Method | Conjugated-xTB QED MAE (↓) | Conjugated-xTB LogP MAE (↓) | Conjugated-xTB GenEff(%)(↑) |
|---|---|---|---|---|
| Chemlactica-125M | Baseline | **0.119** | 20.255 | 60.09 |
| | +CAST | 0.243 | **12.650** | **91.35** |
| | Full SFT | 0.191 | 12.433 | 95.52 |
| | LoRA SFT | 0.147 | 8.941 | 85.53 |
| Chemlactica-1.3B | Baseline | **0.095** | **15.947** | 89.95 |
| | +CAST | 0.106 | 21.279 | **96.40** |
| | Full SFT | 0.243 | 13.715 | 91.40 |
| | LoRA SFT | 0.094 | 6.335 | 79.51 |
| Chemma-2B | Baseline | 0.132 | **34.717** | 50.69 |
| | +CAST | **0.128** | 55.010 | **50.99** |
| | Full SFT | 0.226 | 13.711 | 91.34 |
| | LoRA SFT | 0.119 | 11.421 | 88.51 |

all metrics, even on the Qwen3-4B model. While our primary goal is not to outperform full supervised fine-tuning (full SFT), we include both SFT and LoRA as strong baselines to contextualize the performance of CAST. Compared to LoRA, a widely adopted parameter-efficient fine-tuning method, CAST yields better MAE results in the majority of experiments, as evident in Table 1. Moreover, CAST achieves performance comparable to full SFT, despite not requiring any model fine-tuning. We note that among these proposed methods, with the exception of ActAdd, CAST uses the least amount of trainable parameters, as shown in Table 1. These findings underscore the effectiveness, efficiency, and practicality of CAST as an alternative to parameter-efficient methods.

## 4.2 How well does CAST perform on out-of-distribution data?

To evaluate the robustness of our method under out-of-distribution (OOD) conditions, we conduct experiments in three settings: single OOD property, multiple OOD properties, and Conjugated-xTB properties. For the single property setting, we follow the protocol established in Jolicoeur-Martineau et al. (2024), conditioning on specific values such as QED of 0.1778 or a LogP of 8.1940. In the multiple properties settings, we assess all possible combinations of these single OOD property values, resulting in four OOD combinations. For both settings, we generate 2000 candidate molecules per test case and report aggregated metrics, including Best100MAE and generative efficiency. In addition, to further assess generalization, we evaluate our method on the Conjugated-xTB dataset.

Table 2 presents results for single OOD property target experiments. *Across all models and property targets, CAST outperforms base LLM in almost all cases, while achieving comparable performance otherwise.* Notably, CAST also remains competitive with both LoRA SFT and full SFT, outperforming them in the majority of scenarios. Another significant finding is that CAST consistently and substantially surpasses all baseline models in terms of generative efficiency, achieving the highest proportion of valid and unique molecules across all settings.

These results indicate that CAST provides robust property control in out-of-distribution regimes, without compromising molecular diversity. *For the multiple OOD properties setting (results in Tables 3, 8–10), which involves combinations of the single property targets, we observe that CAST largely outperforms the base models across all combinations, both in terms of Best100MAE and generative efficiency.* Furthermore, CAST remains competitive with the fine-tuned methods in terms of QED alignment, though we note that fine-tuned models achieve better fidelity to LogP property targets. Additionally, we observe that for low and high LogP targets, Chemma-2B struggles to generate proper molecules, resulting in very low generative efficiency. Nonetheless, we also validate that CAST maintains **molecular diversity** when looking at the **number of unique Murcko scaffolds** across all generated SMILES for a given target.

In order to test the robustness of our approach, we also evaluate on the Conjugated-xTB dataset. In this setting, we sample 100 property combinations and generate 100 candidate molecules for each, reporting MAEs and GenEff averaged across all samples. As shown in Table 4, CAST closely matches or exceeds the base LLM in terms of MAEs, while consistently generating with much higher generative efficiency. Interestingly, full SFT struggles to outperform base models on this benchmark, highlighting the significant challenge posed by the Conjugated-xTB dataset.

Table 5: Novelty and synthesizability for in-distribution datasets (2000 samples each). *In PC.* shows the percentage of *valid* generated molecules that are in the PubChem database. We average the SA score (SA sc.) for all the *valid* generated molecules.

| Base LLM | Method | Zinc | | | Moses | | | ChemBL | | |
|---|---|---|---|---|---|---|---|---|---|---|
| | | Valid(%) | In PC. (%) | SA sc. ($\downarrow$) | Valid(%) | In PC. (%) | SA sc. ($\downarrow$) | Valid(%) | In PC. (%) | SA sc. ($\downarrow$) |
| **Chemlactica-** | Baseline | 99.70 | 50.95 | 2.84 | 99.90 | 59.81 | 2.79 | 99.30 | 33.03 | 3.07 |
| **125M** | +CAST | 99.45 | 30.92 | 2.24 | 99.05 | 91.97 | 2.28 | 90.15 | 73.66 | 3.07 |
| **Chemlactica-** | Baseline | 97.25 | 58.10 | 2.59 | 99.95 | 63.33 | 2.50 | 92.00 | 48.09 | 2.83 |
| **1.3B** | +CAST | 96.70 | 95.08 | 2.63 | 100 | 97.75 | 2.55 | 98.40 | 73.68 | 4.07 |
| **Chemma-2B** | Baseline | 100 | 52.80 | 4.74 | 100 | 38.05 | 4.52 | 96.95 | 61.84 | 5.27 |
| | +CAST | 98.65 | 97.77 | 3.32 | 99.85 | 98.65 | 2.61 | 91.00 | 85.99 | 4.26 |

Table 6: Novelty and synthesizability for the OOD setup of Table 3. The In PC. and SA sc. are computed similarly as described in Table 5.

| Base LLM | Method | QED: 01778, LogP: 8.1940 | | |
|---|---|---|---|---|
| | | Valid(%) | In PC. (%) | SA sc. ($\downarrow$) |
| **Chemlactica-125M** | Baseline | 71.75 | 0.42 | 9.18 |
| | +CAST | 90.70 | 3.33 | 6.63 |
| **Chemlactica-1.3B** | Baseline | 86.40 | 7.06 | 6.33 |
| | +CAST | 90.40 | 5.03 | 6.28 |
| **Chemma-2B** | Baseline | 55.80 | 14.87 | 6.20 |
| | +CAST | 60.10 | 16.57 | 6.13 |

Table 7: Maximum and Minimum MAE Variation in the observed property values due to steering strength - Zinc

| Base LLM | Steered | Zinc | | | |
|---|---|---|---|---|---|
| | | QED MAE | | LogP MAE | |
| | | Min. | Max. | Min. | Max. |
| **Chemlactica-** | QED | 0.0201 | 0.4998 | 0.2703 | 9.9348 |
| **125M** | LogP | 0.0267 | 0.6190 | 0.3301 | 28.4044 |
| **Chemlactica-** | QED | 0.0329 | 0.5912 | 0.3760 | 31.8766 |
| **1.3B** | LogP | 0.0326 | 0.5907 | 0.3828 | 31.3571 |

## 4.3 QUALITATIVE ANALYSIS OF CAST

Sections 4.1 and 4.2 demonstrated the superior empirical performance of our proposed CAST method on both in-distribution and out-of-distribution datasets. In this subsection, we take a more qualitative look at the molecules generated by CAST in both setups. Specifically, we focus on two central themes: **novelty** and **synthesizability**. Since our base LLMs are pre-trained on data from PubChem (Kim et al., 2024), we evaluate novelty as whether the generated molecule is a part of the PubChem database. Furthermore, we characterize the molecule's synthesizability using the RD-Kit's **synthetic accessibility score** (SA score) (Ertl & Schuffenhauer, 2009). The SA score ranges between 1 and 10, with 1 being easiest and 10 being hardest to synthesize.

Tables 5 and 6 display the qualitative results of the generated molecules in both in-distribution and out-of-distribution setups. It can be observed that for in-distribution settings, the proposed CAST method tends to recreate more molecules from the PubChem database as compared to the base LLM, thus having similar or better SA scores across all models and all datasets. Combined with its improved MAE values (see Table 1), the concept-based steering guides the LLM generation toward both qualitatively and quantitatively better molecules. It is imperative to note here that for the conditional generation task in this work, our primary focus is on generating valid molecules that obey the desired property values and not on novelty. Similarly, Table 6 shows that while both CAST and base LLM generate more novel molecules in the OOD setting, CAST still outputs relatively more valid molecules from PubChem with better SA scores, especially with Chemlactica-125M, where the SA score is significantly better. *Therefore, CAST improves upon the baselines quantitatively and also provides gains in qualitative terms of synthetic accessibility.*

## 4.4 STEERING STRENGTH VS PROPERTY VALUES

Previous sections have shown the superiority of CAST in generating quantitatively and qualitatively better molecules conditioned on property values. However, the remaining question is how much control this steering has over generated molecules: **does it provide only minor corrections, or can it significantly alter the observed properties?** We answer the question with the following two experimental settings.

**#1: Varying Steering Strength** Here, we vary the steering strength for only one property at a time (say, property $j$), keeping the $c_i$ fixed for all other properties $i \neq j$. Specifically, given the input property values, we perform a forward pass through the LLM and obtain the concept values $c_i, \forall i$. Now, we obtain $\mathbf{z}'_j$ by varying $c_j$ in the range of $[0, 1]$ in increments of 0.1 and calculate the

$\mathbf{z}_{new} = \mathbf{z}'_j + \Sigma_{i \neq j} \mathbf{z}_i$ for each $\mathbf{z}'_j$. This $\mathbf{z}_{new}$ is added to LLM's hidden representation $\mathbf{h}$, and property values are measured for the generated SMILES. From all these property observations corresponding to all $\mathbf{z}_{new}$ for a given sample, we note each property's maximum and minimum MAE. We repeat this procedure for the entire set of samples and report the average max and min MAE.

Table 7 contains the average min and max MAE results on Zinc. It can be observed that varying the steering strength of either QED or LogP can cause significant changes in the observed values for both properties. Take, for example, QED, which ranges between 0 and 1; the max MAE usually is close to 0.5 or 0.6 across all settings, implying that through the steering strength variable, CAST provides enough flexibility to cover a wide range of QED values in its generation.

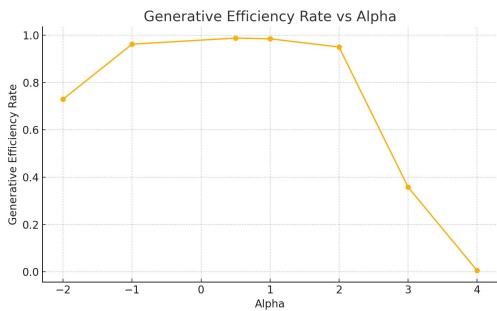

Figure 2: Generative efficiency at varying steering strengths for high QED and high LogP targets.

**#2: Abnormal Steering Strength** Here, we intend to investigate the behavior of LLM when steered with abnormal steering strength values. In particular, we modify the equation for $\tilde{\mathbf{h}} = \mathbf{h} + \mathbf{z}$ as $\tilde{\mathbf{h}} = \mathbf{h} + \alpha\mathbf{z}$ where we vary $\alpha$ in the range of $[-2, 4]$. In the interest of space, we only include results on the OOD setting with high target value for both QED and LogP, and the GenEff metric (Fig. 2). We report the Best100MAE metric in Figure 5 in the appendix.

Figure 2 portrays the variation of the GenEff metric as we vary the values of $\alpha$. It is evident from the plot that the metric begins to deteriorate significantly as the steering strength goes above 2 or below $-1$, symbolizing that abnormally high positive or negative steering strength values can break the model, leading it to generate invalid molecules. Therefore, we can conclude that the steering strength does not behave as an unbounded control parameter; its effect is constrained to a finite range, beyond which the influence either saturates or becomes unstable.

## 5    CONCLUSION AND FUTURE WORKS

In this work, we introduced Concept-based Activation STeering (CAST), a novel framework for conditional molecular generation with large language models. By leveraging a concept bottleneck model, CAST enables direct and precise control over multiple molecular properties without requiring fine-tuning of the base model. Extensive experiments across in-distribution and out-of-distribution benchmarks, including TDC and Conjugated-xTB datasets, demonstrate that CAST achieves strong property alignment and superior generative efficiency compared to the existing activation steering approach. Our results further show that CAST is robust to challenging property targets, offering a practical and flexible alternative to traditional fine-tuning methods. In addition, CAST is able to achieve such performance without sacrificing molecular diversity and quality.

Despite the promising results, we highlight some limitations that point toward important directions for future research. First, the current study focuses on controlling a limited set of continuous molecular properties (QED and LogP), and it remains to be seen how well CAST scales to scenarios requiring control over a larger number of properties. Second, while QED and LogP serve as standard benchmarks, they are relatively straightforward compared to more challenging properties such as HOMO-LUMO gaps or excitation energies, which are highly relevant for real-world drug design. Additionally, CAST currently steers properties at the scalar level but does not explicitly enforce structural or substructural constraints (such as scaffold retention or functional group presence), which are often crucial in practical applications. The framework is also tailored to continuous properties and may require adaptation for categorical, discrete, or graph-structured attributes. Another promising direction is extending CAST from conditional molecular generation to molecule optimization, where the task is to iteratively modify a given starting molecule to achieve desired target properties.

## 6 REPRODUCIBILITY STATEMENT

All experiments and results presented in this paper can be fully reproduced using the code and instructions available at the following anonymous repository: `https://anonymous.4open.science/r/CAST-6BF1`. The repository includes training scripts, evaluation pipelines, and configuration files necessary to replicate our findings.

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

# A  APPENDIX

## A.1  ETHICAL ASSESSMENT

Our work proposes CAST, an activation steering approach for conditional molecular generation. At its backbone, CAST still relies on an LLM which can have implications in critical domains like chemistry and drug discovery. LLMs tend to generate or hallucinate molecules without encoded safety constraints. Therefore, it is recommended to conduct a manual safety check or enforce safety constraints in the architecture itself when using the system for practical real-world situations.

## A.2  USE OF LLMS

Large Language Models (LLMs) were used solely for writing refinement such as grammar and syntax improvements.

## A.3 OOD RESULTS

We further present additional results for the out-of-distribution (OOD) setups. Specifically, Tables 8, 9, and 10 represent a comparison of CAST against baselines for different out-of-distribution properties combinations, such as low QED and low LogP, for example. As shown across these tables, CAST is able to outperform the base LLM most of the time in terms of property alignment and generative efficiency. In addition, we observe that CAST is also able to consistently outperform full supervised-tuning (SFT) models. While we see that LoRA is able to achieve the best performance overall, CAST is able to match its numbers very closely in terms of QED Best100MAE and even better generative efficiency in most cases. We note that Chemma-2B seems to show abnormally low numbers of generative efficiency for low LogP targets, which indicates to us that the model itself has gaps for generating molecules for these specific values. In brief, these numbers further support our claims on the robustness and efficiency of CAST on out-of-distribution input properties.

| Base LLM | Method | Best100MAE(↓) | | GenEff (%) (↑) |
| | | QED | LogP | |
|---|---|---|---|---|
| **Chemlactica-125M** | Baseline | 0.691 | 0.812 | 82.9 |
| | +CAST | **0.440** | **0.741** | **98.5** |
| | Full SFT | 0.614 | 2.132 | 97.2 |
| | LoRA SFT | 0.651 | 0.254 | 85.4 |
| **Chemlactica-1.3B** | Baseline | 0.510 | **0.346** | 95.8 |
| | +CAST | **0.488** | 0.347 | **96.4** |
| | Full SFT | 0.518 | 5.015 | 95.8 |
| | LoRA SFT | 0.661 | 0.065 | 92.5 |
| **Chemma-2B** | Baseline | 0.591 | **1.059** | **49.3** |
| | +CAST | **0.585** | 1.316 | 48.3 |
| | Full SFT | 0.604 | 4.379 | 96.9 |
| | LoRA SFT | 0.565 | 0.308 | 92.1 |

Table 8: Results for property combination: QED Target 1.2861, LogP Target 8.1940

| Base LLM | Method | Best100MAE(↓) | | GenEff (%) (↑) |
| | | QED | LogP | |
|---|---|---|---|---|
| **Chemlactica-125M** | Baseline | 0.0313 | 6.950 | 72.7 |
| | +CAST | **0.0256** | **2.523** | **89.5** |
| | Full SFT | 0.0122 | 0.029 | 76.5 |
| | LoRA SFT | 0.0108 | 0.085 | 82.2 |
| **Chemlactica-1.3B** | Baseline | 0.0108 | 1.794 | 87.1 |
| | +CAST | **0.0089** | **1.373** | **88.8** |
| | Full SFT | 0.0199 | 0.044 | 84.1 |
| | LoRA SFT | 0.0060 | 0.054 | 91.0 |
| **Chemma-2B** | Baseline | 0.0293 | **1.836** | 24.0 |
| | +CAST | **0.0291** | 1.851 | **26.0** |
| | Full SFT | 0.0056 | 0.033 | 84.2 |
| | LoRA SFT | 0.0090 | 0.220 | 64.0 |

Table 9: Results for property combination: QED Target 0.1778, LogP Target -3.2810

## A.4 QUALITATIVE ANALYSIS

This section further presents results on the qualitative assessments of CAST. We first show in tables 11 and 12 the minimum and maximum MAE variation observed due to changes in steering strength. Similarly to the conclusions we mention in the main sections, we can clearly observe that by varying steering strength, CAST allows us to explore a wide range of property values. We also add Figure 5 that shows the relationship between the overall steering strength and MAE. The graph shows that as

| Base LLM | Method | Best100MAE($\downarrow$) QED | Best100MAE($\downarrow$) LogP | GenEff (%) ($\uparrow$) |
|---|---|---|---|---|
| **Chemlactica-125M** | Baseline | 0.734 | 2.572 | 81.0 |
| | +CAST | **0.446** | **1.357** | **98.4** |
| | Full SFT | 0.734 | 0.027 | 76.8 |
| | LoRA SFT | 0.602 | 0.096 | 81.6 |
| **Chemlactica-1.3B** | Baseline | 0.496 | 1.467 | 95.3 |
| | +CAST | **0.494** | **1.238** | **96.6** |
| | Full SFT | 0.625 | 0.043 | 85.6 |
| | LoRA SFT | 0.604 | 0.054 | 93.5 |
| **Chemma-2B** | Baseline | **0.685** | **1.365** | **26.6** |
| | +CAST | 0.693 | 1.524 | 26.0 |
| | Full SFT | 0.683 | 0.032 | 75.1 |
| | LoRA SFT | 0.582 | 0.120 | 90.0 |

Table 10: Results for property combination: QED Target 1.2861, LogP Target -3.2810

we increase the steering strength, we have an increasing trend of misalignment in property values. We also see that at one point ($\alpha = 2$), the steering breaks the generation where we see a high deviation in terms of MAE. This phenomenon could also be seen in Figure 2, where it happens on generative efficiency.

Tables 13, 14 and 15 presents qualitative metrics on a OOD setting. Specifically, it presents the proportion of molecules found in PubChem and the average SA score at different OOD property value combinations. Similar to what we describe above, we see that CAST pushes the generation toward molecules that are found in PubChem. This, in turn, improves its SA scores as molecules found in PubChem are known, valid and good quality molecules. This further shows that CAST is able to generate high quality molecules.

| Base LLM | Steered | Moses QED MAE Min. | Max. | LogP MAE Min. | Max. |
|---|---|---|---|---|---|
| **Chemlactica-125M** | QED | 0.0158 | 0.5006 | 0.2022 | 7.6618 |
| | LogP | 0.0212 | 0.6443 | 0.2326 | 24.6594 |
| **Chemlactica-1.3B** | QED | 0.0341 | 0.6356 | 0.3115 | 28.6960 |
| | LogP | 0.0369 | 0.6309 | 0.3201 | 28.9727 |

Table 11: Maximum and Minimum MAE Variation in the observed property values due to steering strength - Moses Dataset.

| Base LLM | Steered | ChemBL QED MAE Min. | Max. | LogP MAE Min. | Max. |
|---|---|---|---|---|---|
| **Chemlactica-125M** | QED | 0.0321 | 0.4855 | 0.5075 | 17.3955 |
| | LogP | 0.0422 | 0.5231 | 0.7082 | 39.5629 |
| **Chemlactica-1.3B** | QED | 0.0330 | 0.4706 | 0.8700 | 46.3020 |
| | LogP | 0.0327 | 0.4711 | 0.8202 | 46.3962 |

Table 12: Maximum and Minimum MAE Variation in the observed property values due to steering strength - ChemBL Dataset.

## A.5 EXAMPLES OF GENERATED MOLECULES

Figure 3 shows some example drawings of the molecules generated by our proposed CAST method. It can be observed that the method creates molecules of varying complexity (different branching, dif-

| Base LLM | Method | QED: 1.2861, LogP: 8.1940 | | |
|---|---|---|---|---|
| | | Valid(%) | In PC. (%) | SA sc. (↓) |
| **Chemlactica-125M** | Baseline | 82.90 | 2.33 | 7.53 |
| | +CAST | 98.60 | 10.76 | 3.34 |
| **Chemlactica-1.3B** | Baseline | 95.80 | 14.16 | 4.53 |
| | +CAST | 96.35 | 16.63 | 4.50 |
| **Chemma-2B** | Baseline | 50.05 | 23.00 | 5.13 |
| | +CAST | 48.80 | 22.23 | 5.14 |

Table 13: Novelty and synthesizability for the OOD setup of Table 8. The In PC. and SA sc. are computed similarly as described in Table 5.

| Base LLM | Method | QED: 01778, LogP: -3.2810 | | |
|---|---|---|---|---|
| | | Valid(%) | In PC. (%) | SA sc. (↓) |
| **Chemlactica-125M** | Baseline | 73.15 | 0.07 | 9.22 |
| | +CAST | 89.45 | 1.75 | 6.88 |
| **Chemlactica-1.3B** | Baseline | 87.10 | 6.48 | 6.26 |
| | +CAST | 88.80 | 6.77 | 6.29 |
| **Chemma-2B** | Baseline | 25.10 | 24.79 | 6.54 |
| | +CAST | 27.25 | 22.24 | 6.52 |

Table 14: Novelty and synthesizability for the OOD setup of Table 9. The In PC. and SA sc. are computed similarly as described in Table 5.

ferent ring structures) with a variety of functional groups and atoms other than standard carbon and hydrogen atoms. These drawings also confirm that the generated SMILES are not only syntactically right, but also represent chemically coherent and plausible molecules.

| Base LLM | Method | QED: 1.2861, LogP: -3.2810 | | |
|---|---|---|---|---|
| | | Valid(%) | In PC. (%) | SA sc. (↓) |
| **Chemlactica-125M** | Baseline | 81.30 | 2.14 | 7.61 |
| | +CAST | 98.40 | 11.30 | 3.40 |
| **Chemlactica-1.3B** | Baseline | 95.30 | 16.55 | 4.48 |
| | +CAST | 96.55 | 16.93 | 4.48 |
| **Chemma-2B** | Baseline | 26.85 | 26.62 | 5.52 |
| | +CAST | 26.60 | 23.69 | 5.61 |

Table 15: Novelty and synthesizability for the OOD setup of Table 10. The In PC. and SA sc. are computed similarly as described in Table 5.

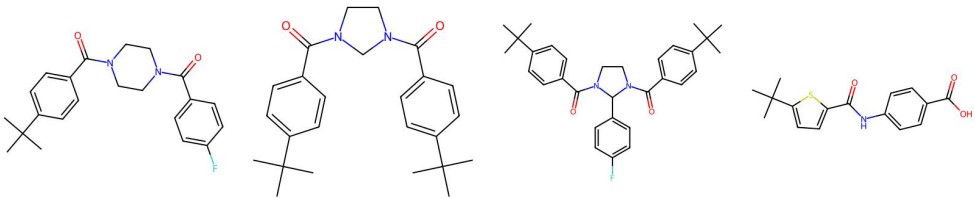

Figure 3: Drawings of SMILES strings of Molecules generated by CAST

Example 1:

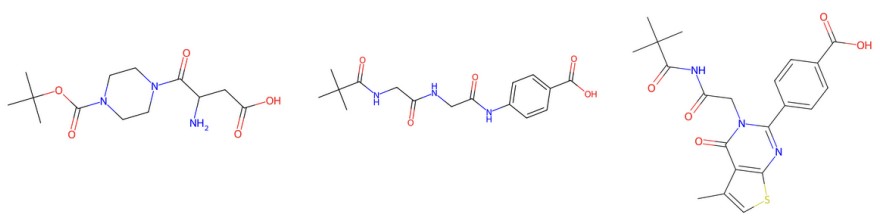

Example 2:

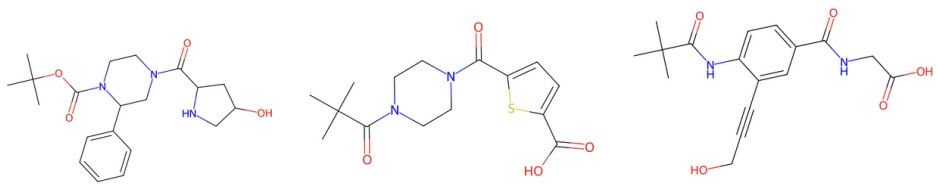

Figure 4: Drawings of SMILES strings of molecules generated by CAST from the same input but with varying steering strengths. From left to right, the steering strength are 1, 2 and 3.

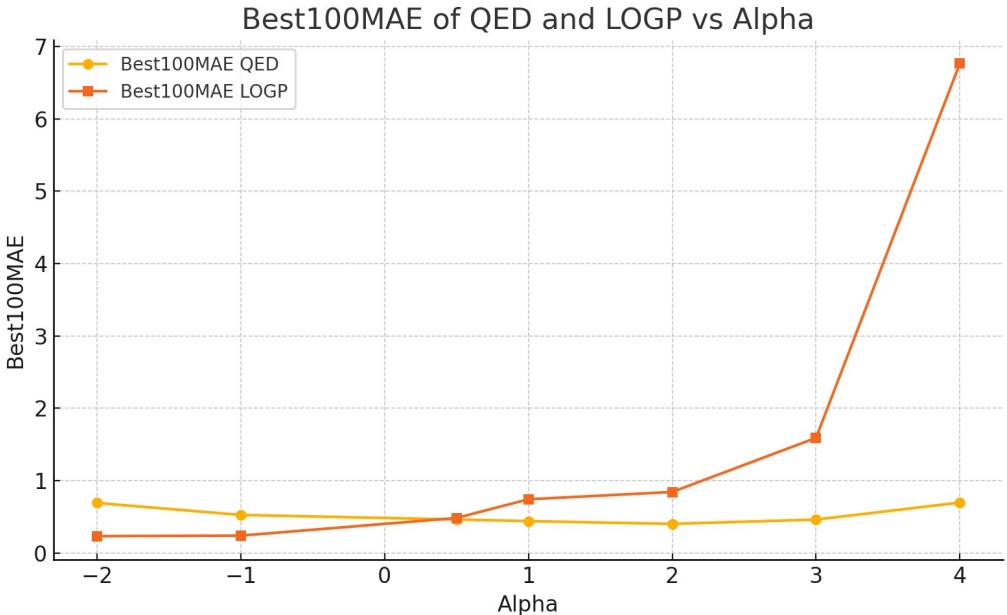

Figure 5: Best100MAEs at varying steering strengths for high QED and high LogP targets

