# OpenReview forum: "Concept-Based Steering of LLMs for Conditional Molecular Generation"
_ICLR.cc/2026/Conference — Submitted to ICLR 2026_

### Official Review · Reviewer_guuH · 2025-10-27

**Soundness:** 2
**Presentation:** 1
**Contribution:** 1
**Rating:** 2
**Confidence:** 4

**Summary:**

The authors introduce concept-based activation steering (CAST) for conditional molecular generation with LLMs.
CAST enables precise control over multiple molecular properties without requiring model fine-tuning.
Extensive experiments on in-distribution and out-of-distribution datasets showcase CAST's superior property alignment and generative efficiency compared to existing methods.
Results demonstrate that CAST consistently outperforms base LLMs and even competitive fine-tuned models in most scenarios.
The method maintains molecular diversity and quality while achieving strong property alignment.
Qualitative assessments reveal that CAST generates molecules with higher novelty and synthesizability, improving the quality of generated molecules.
The method's robustness and efficiency are evident across various datasets and out-of-distribution scenarios.

**Strengths:**

- The paper transfers the idea of activation steering from general NLP tasks to conditional molecular generation, employing a Concept Bottleneck Model (CBM) to learn interpretable property directions (QED and LogP) for dynamic activation modulation. The paper clearly points out the limitations of traditional methods that rely on handcrafted contrastive prompts and fixed steering strengths, whereas CAST introduces a CBM-based, automated, and continuous way to generate property directions, with a logically coherent and well-structured design.

- The experiments cover multiple scenarios within the TDC datasets (ZINC, MOSES, and ChEMBL) and OOD settings, comparing CAST with Baseline, ActAdd, LoRA, and full SFT models, and reporting metrics such as MAE, Validity, and GenEff. CAST outperforms the baselines in most tables while using far fewer parameters than LoRA or full SFT, though in some specific settings SFT or LoRA achieve better results.

**Weaknesses:**

- Although the paper introduces a CBM to automatically generate steering vectors, the overall framework essentially remains a standard activation addition paradigm. Compared with existing approaches such as Activation Addition, Contrastive Steering, and Representation Engineering, the contribution is mainly at the application level, lacking new theoretical insights or substantial algorithmic innovation.

- While the CAST framework indeed extracts hidden states from intermediate layers and passes them through the CBM to generate concept vectors that are re-injected into the residual stream, the choice of layer position is an empirical design decision. The authors justify it only by citing prior work claiming that “intermediate layers retain rich semantic information,” without providing mechanistic explanations or conducting layer-wise ablation to validate this choice or its uniqueness.

- The paper does not justify the selection of QED and LogP as target properties, nor explain their chemical or pharmacological relevance to generative controllability.”

- The work lacks a discussion on the causes of OOD performance degradation, and the OOD task design itself is relatively simplistic.

**Questions:**

Please see weaknesses.

---

> ### Author Response · Authors · 2025-11-14
>
> We thank the reviewer for the assessment and constructive feedback. We clarify several points and address the questions below.
>
> **Contribution beyond standard application**
>
> While CAST builds on the activation-steering paradigm, its contribution is not merely an application of existing methods. Prior approaches require:
>
> 1. manually crafted contrastive prompts (positive/negative examples) for each concept,
> 2. computing one separate steering vector per concepts
> 3. fixed or fine-tuned steering strengths
>
> In contrast, CAST introduces a continuous, data-driven mechanism for deriving steering vectors via a Concept Bottleneck Model (CBM). The CBM learns property-conditioned directions directly from hidden states, enabling:
>
> 1. dynamic steering for any target value
> 2. multi-property control without requiring 2x datasets per concept
>
> This eliminates the need for handcrafted contrastive datasets and manual steering-strength tuning, which constitute the primary limitations of prior steering methods. This adds substantive methodological novelty, not just application-level adaptation.
>
> **Justification for intermediate-layer steering**
>
> We thank the reviewer to point out how we chose layer into which the CBM's vector is added. We will provide the ablation study numbers as soon as we have them.
>
> **Choice of QED and LogP as target properties**
>
> We thank the reviewer for pointing out the lack of justification behind why we chose QED and LogP as target properties. We omitted this as QED and LogP are standard, widely used benchmarks in molecular generation and property  literature for their ease of computation. We highlight their relevance:
>
> 1. QED captures drug-likeness
> 2. LogP captures lipophilicity and is central synthesis constraints
>
> We will clarify the rationale and motivation behind their choice as target properties in our revised revision.
>
> **Clarification on OOD setting**
>
> We thank the reviewer for raising the question regarding OOD performance degradation. To clarify, in our paper we clearly state that the degradation primarily stems from the inherent difficulty of generating molecules whose target properties lie outside the distribution of the training data. We specifically study two challenging OOD settings:
>
> 1. Single-property and multi-property extreme targets: Here, we explicitly condition on extreme or highly rare property values that are scarcely represented in known chemical space.
> 2. The Conjugated-xTB dataset: We state the difficult nature of this dataset in Section 4 (lines 245-248).
>
> These experiments were designed to probe robustness under genuine distributional shifts. We will expand the description of the OOD motivation and experimental setup in the revised manuscript to make this clearer.
>
>
> We hope these clarifications address the mentioned concerns and make the contribution of CAST more apparent.

---

> > ### Author Response · Authors · 2025-11-16
> >
> > We performed a layer-wise ablation on Chemlactica-125M a 12-layer model and found that intermediate layers yield substantially better controllability than early or late layers. For space efficiency, we refer the reviewer to the table provided in our response to Reviewer 8nFv, where these results are reported in full. The findings support our design choice to intervene at an intermediate layer.
> >
> > We hope this helps clarify our methodology.

---

> ### Author Response · Authors · 2025-11-19
> **Kind reminder about rebuttal**
>
> Dear Reviewer,
>
> Thank you for acknowledging the extent of our experiments. We wanted to kindly follow up on our earlier responses to your concerns. If you have any further questions or feedback, we would be happy to clarify. We hope that our clarifications can positively inform your assessment of the contribution and novelty of our work.
>
> Thank you again for your time and consideration.
>
> Best regards.

---

### Official Review · Reviewer_8nFv · 2025-10-29

**Soundness:** 1
**Presentation:** 3
**Contribution:** 2
**Rating:** 4
**Confidence:** 4

**Summary:**

This paper proposes CAST, a concept-based activation steering approach for molecular LLMs that aims to control continuous properties without fine-tuning the base LLM. The authors train a small concept bottleneck module on top of a frozen decoder-only LLM to map its hidden states to property-specific directions, which are later injected at an intermediate layer during decoding. On ZINC/MOSES/ChEMBL and OOD targets (incl. Conjugated-xTB), CAST reduces MAE versus frozen baselines, where the authors also argue that CAST also enables high validity, novelty, and synthesizability.

**Strengths:**

- The authors present a method that introduces a simple auxiliary network to learn appropriate hidden features for steering, without any direct training or modification of the base LLM.

- They conduct experiments across diverse application scenarios and, in addition to standard baselines, include both full fine-tuning and LoRA-based fine-tuning, enabling a thorough performance comparison.

**Weaknesses:**

- The overall performance of CAST still lacks adequate justification: the CBM module’s outputs are trained solely with a separate MLP-based property regression loss, and the training pipeline does not include any end-to-end objective that evaluates the effect of injecting these features into the LLM’s hidden states during generation. Consequently, there is no principled guarantee that adding these features will improve property control or overall generation quality.

- Similarly, the expectation that validity, novelty, and synthesizability would improve is insufficiently supported; indeed, the tendencies reported in Tables 5–6 do not consistently substantiate these claims and, in some cases, appear neutral or mixed with respect to such improvements.

**Questions:**

- Considering that the target properties can be correlated (e.g., QED partially depends on LogP), how essential is the orthogonality loss? Could the authors quantify the performance sensitivity to including versus removing this term, and report whether any observed differences are significant?

- Is there a justification(e.g., an ablation or comparative study) for the choice of the intermediate layer at which CBM features are injected? How does performance vary across candidate layers and depths, and is there a principled criterion guiding this selection?

---

> ### Author Response · Authors · 2025-11-14
>
> We thank the reviewer for the detailed feedback and constructive questions. We appreciate the time taken and address each point below.
>
> **Clarification and Motivation of the Training Design**
>
>
> Thank you for pointing out an important point regarding end-to-end guarantees of our model training and design. Our design choice was deliberate: using only a supervised property-regression objective on the CBM is what allows CAST to remain stable and efficient. We would like to emphasize that the key motivation of our approach was to build upon the activation steering literature where the larger backbone model is kept frozen for efficiency purposes.
>
> Nevertheless, while we agree that the idea of adding an objective that evaluates generated molecules could be promising, the following challenges might arise in the suggested approach:
> 1. Molecular properties calculated from generated SMILES are non-differentiable, thus requiring reinforcement learning or approximate gradient estimators.
> 2. the reward signals are sparse and noisy due to invalid SMILES
> 3. the training loop becomes computationally expensive due to repeated oracle calls for property predictions, and
> 4. the training becomes unstable due to challenges in design a proper reward signal (property alignment, validity, etc.)
>
> CAST was explicitly designed to avoid these pitfalls while still enabling strong control. Thus, we show empirically that the CBM-learned directions can reliably steer the LLM toward the desired property values without compromising qualitative metrics (validity, uniqueness, etc.), confirming the soundness of this simpler and more efficient formulation of learning steering vectors.
>
> **Justification for intermediate-layer steering**
>
> We thank the reviewer to point out how we decided on the layer into which the CBM's vector is applied. Our choice follows extensive prior works on activation steering and mechanistic interpretability in language models, which consistently found that intermediate layers contain the most semantically rich and controllable representations. We will also provide ablation study numbers for this as soon as possible.
>
> **Clarification on novelty and Tables 5-6**
>
> We thank the reviewer for the thoughtful question regarding the expectation of novelty, validity and synthesizability improvements. We clearly state in Section 4.3 (line 415) of our paper that *our primary focus in this work is on generating valid molecules that obey the desired property values and not on novelty*. We also want to clarify the two notions of novelty used in the paper:
>
> 1. In Tables 5-6, the "In PubChem" column measures *external* novelty (i.e. whether the molecule exists in the broader PubChem database).
> 2. The GenEff metric, used throughout the paper, captures *internal* novelty within the generated batch alongside validity and uniqueness.
>
> **Importance of the orthogonality loss**
>
> We thank the reviewer for the note on the sensitivity of the orthogonality loss. We performed this ablation and observed only minor changes across all metrics when removing the orthogonality loss. Our architecture already uses separate MLP heads per property, which provides a strong degree of disentanglement by design. The orthogonality term serves as a regularizer to enhance robustness, especially for scenarios where properties could have stronger correlations.
>
>
> We appreciate the reviewer's feedback and hope that the clarifications and forthcoming additions address the concerns raised.

---

> > ### Author Response · Authors · 2025-11-16
> > **Ablation numbers for intermediate layer choice**
> >
> > To clarify the concern about which layer to steer, we trained CBMs on early and late layers, respectively layer 2 and 10 of Chemlactica-125M of 12 layers. We compare performance to our results (layer 6) across Zinc, Moses and ChEMBL datasets below:
> >
> > **Layer-wise Ablation (12-layer Chemlactica-125M model)**
> >
> > | Dataset | Layer | QED MAE ↓ | LogP MAE ↓ | Validity ↑ |
> > |---------|--------|------------|-------------|-------------|
> > | **ZINC** | 2 | 0.127 | 1.818 | 99.2 |
> > |  | **6 (ours)** | **0.075** | **0.611** | **100** |
> > |  | 10 | 0.124 | 1.736 | 98.8 |
> > | **MOSES** |  2 | 0.154 | **1.011** | 99.1 |
> > |  | **6 (ours)** | **0.078** | 1.232 | **100** |
> > |  |  10 | 0.124 | 1.171 | 99.1 |
> > | **ChEMBL** | 2 | 0.249 | 2.635 | 98.4 |
> > |  | **6 (ours)** | **0.226** | **1.738** | **99.5** |
> > |  | 10 | 0.264 | 2.116 | 98.1 |
> >
> > Across all datasets, the intermediate layer (layer 6) provides the best overall performance. This behaviour aligns with prior steering work. These results empirically support and confirms that intermediate layers provide the most stable and effective steering point.
> >
> > We hope this clarifies our methodology.

---

> ### Author Response · Authors · 2025-11-19
> **Kind reminder about rebuttal**
>
> Dear Reviewer,
>
> Thank you for acknowledging the extensive performance comparison of our work. We wanted to kindly follow up on our earlier responses to your concerns. If you have any further questions or feedback, we would be happy to clarify. We hope that our clarifications can positively inform your assessment of the soundness and contribution of our work.
>
> Thank you again for your time and consideration.
>
> Best regards.

---

> > ### Comment · Reviewer_8nFv · 2025-11-24
> >
> > I appreciate the response from the authors and the additional experimental results, including feature injection location and the interpretation of Novelty and Synthetic Accessibility. That said, I still have several concerns about the soundness of the work:
> >
> > - While I understand the choice of having loss on a separate network (without utilizing LLM end-to-end objective) to be a more 'stable' method, this makes unclear about 'why CAST would work'; how can we guarantee injecting vectors from a different feature space basis $e_i$ into LLM feature makes the LLM output to follow the target property, rather than doing nothing or even being negatively correlated? Or should this only be understood as a 'discovery'?
> >
> > - Considering many chemical properties are inherently not disentangled(e.g., logP and QED), and the rebuttal says the performance difference according to the orthogonality loss $L_{orth}$ is minor, what would be the contribution of $L_{orth}$ in this paper? Although the authors claimed this can serve as a regularizer, this claim hasn't been backed with a theoretical explanation or experimental results yet. Plus, I think the explicit formula for $L_{orth}$ must be described.
> >
> > For now, I would like to maintain the current score according to the issues mentioned above.

---

> > > ### Author Response · Authors · 2025-11-24
> > >
> > > Dear Reviewer,
> > >
> > > Thank you for your response. We address each concern below:
> > >
> > > 1) We agree that CAST, like prior activation steering methods, does not provide a formal guarantee that property alignment will always improve after steering. Rather, it should be understood as a representation discovery framework, where the CBM is trained on the LLM’s own hidden activations to identify directions that correlate with target molecular properties. These directions are therefore not drawn from a separate feature space, but discovered within the LLM’s internal representation itself. In this sense, CAST builds upon insights from activation steering and representation engineering literature by providing a systematic mechanism for discovering property-aligned directions that enable controllable molecular generation.
> > >
> > > 2) The orthogonality term is not essential for performance in our current setting, but it provides a principled regularization that improves stability and interpretability, particularly for more complex multi-property scenarios. In practice, a significant degree of disentanglement is already encouraged by our architectural design, since each property is modeled with a separate MLP head. The orthogonality loss therefore serves as an additional structural bias rather than a necessary component. The orthogonal loss used follows the common cosine similarity formulation:
> > > $$L_{orth} = \frac{e_i^\top e_j}{\lVert e_i \rVert \ \lVert e_j \rVert}$$
> > >
> > > We hope this adds further clarification to the soundness of our work.

---

### Official Review · Reviewer_U2g5 · 2025-11-01

**Soundness:** 3
**Presentation:** 2
**Contribution:** 3
**Rating:** 4
**Confidence:** 2

**Summary:**

The paper introduces Concept-based Activation STeering (CAST), a lightweight and flexible method for controlling large language models (LLMs) to generate molecules with specific properties, such as drug-likeliness (QED) and solubility (LogP). Unlike traditional methods that require costly retraining or fine-tuning of the LLM , CAST keeps the base LLM frozen. It works by training a small Concept Bottleneck Model (CBM) that learns to map the LLM's internal activations to property-specific steering vectors. During inference, these vectors are added back into the LLM's hidden states to "steer" the generation process toward the desired property values. Through extensive experiments, CAST is shown to consistently outperform baseline models and other activation steering methods, achieving strong performance on both in-distribution and out-of-distribution (OOD) generation tasks and demonstrating superior generative efficiency compared to even fine-tuned models.

**Strengths:**

1. **Efficiency and Flexibility**: The primary strength of CAST is that it does not require retraining or fine-tuning the base LLM, whose parameters remain frozen. This makes it a highly efficient and flexible alternative to SFT and LoRA, which require costly retraining for new property distributions.


2. **Novel Steering Mechanism**: The paper successfully improves upon traditional activation steering. By using a CBM to automatically compute property-conditioned steering vectors , CAST avoids the major limitations of prior work, such as the reliance on manually crafted contrastive prompts and fixed steering magnitudes.



3. **Strong OOD Performance**: The method demonstrates remarkable robustness on out-of-distribution (OOD) tasks. In settings where target properties are outside the training distribution, CAST consistently outperforms baseline models and remains highly competitive with fully fine-tuned models.

**Weaknesses:**

1. **Limited Property Control**: The method was only evaluated on two continuous molecular properties, QED and LogP. Though the authors note that QED and LogP are "relatively straightforward" benchmarks, the method's effectiveness on more complex or challenging properties, such as HOMO-LUMO gaps or excitation energies, remains untested.

2. **Lack of Qualitative Examples**: While the paper presents extensive quantitative results that are convincing, they can also be confusing: how this simple steering technique intuitively influences such a complex molecular generation process? To better elucidate the underlying mechanism, could the authors show how a specific molecule's SMILES string or 2D graph varies as they continuously change the steering strength for a single property?

**Questions:**

1. As I am new to the concept of activation steering, I would appreciate a more detailed illustration of the inference-time procedure. Specifically, regarding the concept values $c_i$, are they predicted by the model during inference, with their strength then tuned by the users (i.e., scaled up or down) to find an optimal value? Or, alternatively, are the $c_i$ values entirely user-specified at inference, with the Concept Bottleneck Model (CBM) being completely dropped? My current understanding favors the first interpretation. If this is correct, does it imply that generating a specific molecule requires careful manual tuning of the steering strength to achieve an optimal result? This can be a weakness.


2. See W2.

3. If my understanding in Q1 is correct, I have a follow-up regarding the claim in Experiment 4.4 #1: "Take, for example, QED... the max MAE usually is close to 0.5 or 0.6 across all settings, implying that through the steering strength variable, CAST provides enough flexibility..." Could the authors please elaborate on the use of the term "usually" in this context? I am asking for clarification because, upon reviewing Table 7, some of the reported max MAE values appear to be substantially larger than 0.6. How is the claim reconciled with these larger values in Table 7?


I will consider raising my score if my questions or misunderstandings of this paper are properly addressed.

---

> ### Author Response · Authors · 2025-11-14
>
> We thank the reviewer for the thoughtful and constructive feedback. We appreciate the time spent engaging with our work and are happy to clarify the points raised.
>
> **Limited Property Control**
>
> To our knowledge, our work is the first approach to apply activation steering to directly edit a model’s internal representation for conditional molecular generation. Therefore, we provide a proof a concept with well-studied properties in this work, and leave more complex properties for future work.
>
> **Clarification on inference procedure**
>
> At inference time, the concept values $c_i$ are predicted by the CBM given the LLM's hidden activations. These values are then used with the learned concept embeddings $e_i$ to compute the corresponding steering vector $z_i$, which is added to the layer's activations. We describe this procedure in more detail in Section 3.1 Equation 1 and 4. Importantly, we do not include additional steering strength hyperparameter as the concept values take that role. Thus, there is no manual tuning required. This is a key advantage of our method compared to other traditional steering method since we automate this process.
>
> **Clarification on "usually" and MAE values**
>
> The statement in Section 4.4 #1 ("Take, for example, QED, which ranges between 0 and 1; the max MAE  usually
> is close to 0.5 or 0.6 across all setting") refers specifically to QED MAE values in Table 7. The larger max MAE values in Table 7 correspond to LogP, which has a much broader numeric range.
>
> **Qualitative illustration of steering effects**
>
> We provide in the Appendix (Fig. 4) two qualitative examples of 3 molecules generated with the same input but with varying steering strength (1,2,3). We want to highlight that the concept vectors learned by the CBM are actually a latent representation of the concepts i.e. chemical properties like QED and LogP. Therefore, modifying their respective strength values signifies modifying the extent of that property exhibited by the generated molecule, which may not directly correlate with structural complexity. The significant gap between min and max QED and LogP values in Table 7 further corroborate this. However, we want to note that beyond a steering strength of 3 the model typically collapses and fails to generate valid molecules SMILES which makes it impossible to properly compare.
>
>
> We appreciate the reviewer's feedback and hope that the clarifications and forthcoming additions address the concerns raised.

---

> > ### Comment · Reviewer_U2g5 · 2025-11-27
> >
> > Thank you for your response! Your clarification helped clear up my misunderstanding. However, I remain concerned about the scope of the experiments (e.g., tested property). Additionally, after reading the comments of other reviewers and reviewing related techniques in the field, I am concerned about the novelty of the proposed method. For these reasons, I will maintain my score but with a lower confidence level.

---

> > > ### Author Response · Authors · 2025-11-27
> > >
> > > Dear Reviewer,
> > >
> > > We appreciate your follow-up on our response.
> > >
> > > Regarding the experimental scope, while we agree that we did not test for a large number of molecular properties because this work was a first attempt of using activation steering for molecular generation, the scope of our experiments remain extensive. To properly assess our proposed method, we performed:
> > > - In-distribution experiments on 3 datasets across 4 models
> > > - Out-of-distribution experiments across 3 settings (single property, multi-properties, and with x-TB molecule dataset)
> > > - Qualitative assessment of synthesizability and novelty with molecules from PubChem
> > >
> > >
> > > Regarding the aspect of novelty, our proposed method differ greatly from traditional steering methods. Similar to our response to Reviewer guuH, while CAST builds on the activation-steering paradigm, its contribution is not merely an application of existing methods. Our approach tackles the main caveats of traditional activation steering:
> > > - manually crafted contrastive prompts (positive/negative examples) for each concept,
> > > - computing one separate steering vector per concepts
> > > - fixed or fine-tuned steering strengths
> > >
> > > CAST introduces a new approach that allows for **dynamic** and **multi-property** steering as well as having data-driven **steering strength adaptation**. To our knowledge, this is the **first** work to transform activation steering from a static, handcrafted process into a learned, continuous control mechanism over internal representations. This shifts steering from being a post-hoc manipulation technique to a learned representational control model, enabling broader generalization, scalability, and multi-concept compositionality in ways not supported by prior methods.
> > >
> > > We hope this clarifies concerns about novelty and clarifies the extent of the experiments performed in our work.

---

> ### Author Response · Authors · 2025-11-19
> **Kind reminder about rebuttal**
>
> Dear Reviewer,
>
> Thank you for acknowledging the strengths and novelty behind our work. We wanted to kindly follow up on our earlier responses to your concerns. If you have any further questions or feedback, we would be happy to clarify. We hope that our clarifications can positively inform your assessment of our work.
>
> Thank you again for your time and consideration.
>
> Best regards.

---

### Author Response · Authors · 2025-11-30
**Summary of Reviewer Feedback and Our Revisions**

We thank the reviewers for their insightful feedback during this period of time. We appreciate that they recognize the method proposed in our work as *"novel"*, *efficient"* and *"flexible"* while demonstrating *"strong"* performance over different benchmarks (reviewer U2g5); that we conduct a *"thorough performance comparison"* across diverse application scenarios (reviewer 8nFv); that our work (CAST) introduces a *"logically coherent and well-structured design"* way that tackles the limitations of traditional methods of activation steering (reviewer guuH).

**We also acknowledge the critiques raised by the reviewers.** Substantial ones among them were:

- Clarification and motivation of the training design;
- Justification for intermediate-layer steering
- Inclusion of more qualitative examples of how steering affects molecular generation

**Our main follow-up to these comments are as follow:**

- We provide a detailed justification for our training design, specifically our choice to use a supervised property-regression objective on the CBM without end-to-end LLM training. We outline the practical challenges involved and explain why our approach is aligned with and motivated by prior activation-steering literature.
- We conduct a layer-wise ablation study to support our choice of steering at an intermediate layer. The results confirm and strengthen our initial design decision.
- We perform an ablation on steering strength and provide additional qualitative examples of generated molecules. We further clarify that adjusting steering strength modulates the degree to which a property is expressed in the generated molecule, which does not necessarily correlate with structural complexity.

We also address several additional points raised in the reviews, including our motivation for choosing QED and LogP as target properties, the expected performance degradation in the OOD setting, and clarifications regarding specific evaluation metrics. All corresponding revisions have been incorporated into the manuscript.

---

### Meta-Review · Area_Chair_1yrP · 2025-12-06

**Summary:**

This work proposes a new method leveraging activation engineering to achieve efficient, flexible, and scalable conditional molecular generation. However, during the initial review stage, multiple reviewers raised concerns about the limited evaluation scope—CAST is tested only on two relatively simple properties (QED and LogP), without evidence of generality to broader or more challenging chemical attributes. Reviewers also noted the absence of intuitive qualitative examples demonstrating how steering strength influences generated molecular structures, as well as insufficient clarity regarding the inference-time procedure. More critically, several reviewers highlighted missing empirical justifications for key design choices, including the role of the orthogonality loss, the selection of the intermediate injection layer, and the lack of an end-to-end objective that ensures CBM-generated features truly improve controllability or generation quality.

In the rebuttal, the authors actively addressed many of these concerns by clarifying the rationale behind property choices, providing additional OOD-setting details, and adding layer-wise ablations.

Nevertheless, significant reservations remain among the reviewers regarding the novelty of the proposed method and the substantive contribution of certain loss components. Given these unresolved issues, I recommend rejection.

**Reviewer Concerns:**

Addressed: Clarification and motivation of the training design;OOD setting details; more qualitative examples;


Still outstanding. The novelty of the proposed method and the substantive contribution of certain loss components.

**Reviewer Scores:**

Reviewer 8nFv and Reviewer U2g5 engaged in substantial discussion during the post-rebuttal phase but ultimately leaned toward maintaining their initial scores.

Reviewer guuH did not participate in the discussion.

---

### Decision · Program_Chairs · 2026-01-26

Reject